# Characterization and Genomic Analysis of a Novel Lytic Phage DCp1 against *Clostridium perfringens* Biofilms

**DOI:** 10.3390/ijms24044191

**Published:** 2023-02-20

**Authors:** Zhaohui Tang, Xiaojing Li, Xinwei Wang, Can Zhang, Ling Zou, Huiying Ren, Wenhua Liu

**Affiliations:** College of Veterinary Medicine, Qingdao Agricultural University, Qingdao 266109, China

**Keywords:** *C. perfringens*, phage DCp1, genomic characterization, biological characterization, bacterial biofilm

## Abstract

*Clostridium perfringens* (*C. perfringens*) is one of the foremost pathogens responsible for diarrhea in foals. As antibiotic resistance increases, phages that specifically lyse bacteria are of great interest to us with regard to *C. perfringens*. In this study, a novel *C. perfringens* phage DCp1 was isolated from the sewage of a donkey farm. Phage DCp1 had a non-contractile short tail (40 nm in length) and a regular icosahedral head (46 nm in diameter). Whole-genome sequencing indicated that phage DCp1 had a linear double-stranded DNA genome with a total length of 18,555 bp and a G + C content of 28.2%. A total of 25 ORFs were identified in the genome, 6 of which had been assigned to functional genes, others were annotated to encode hypothetical proteins. The genome of phage DCp1 lacked any tRNA, virulence gene, drug resistance gene, or lysogenic gene. Phylogenetic analysis indicated that phage DCp1 belonged to the family *Guelinviridae*, *Susfortunavirus*. Biofilm assay showed that phage DCp1 was effective in inhibiting the formation of *C. perfringens D22 biofilms*. Phage DCp1 could completely degrade the biofilm after 5 h of interaction. The current study provides some basic information for further research on phage DCp1 and its application.

## 1. Introduction

*C. perfringens* is a sporulated, toxin-producing Gram-positive anaerobe in the environment, and it is an opportunistic pathogen that could cause an imbalance of intestinal flora and induce the development of diseases (such as diarrhea) [1,2]. The *C. perfringens* exotoxins are classified into two categories, major and minor toxins [3]. The major toxins include alpha (CPA), beta (CPB), epsilon (ETX), and iota (ITX) toxins [4], based on which the strains of *C. perfringens* are classified into five types (A–E). *C. perfringens* has been reported to cause Clostridial enteritis in foals and adult horses in many countries [5,6,7]. In the early stages of the disease, foals may have anorexia, diarrhea, depression, and dehydration. Intestinal hypomotility or paralytic ileus may also be present. Clostridial enteritis may also be associated with enterotoxemia, systemic inflammatory response syndrome (SIRS), and sepsis, and it is considered a leading cause of death [3]. The onset and progression of clostridial enteritis are rapid and can result in extremely high mortality in foals. Clostridial enteritis has seriously affected the development of the donkey industry in the world. Although *C. perfringens* can be controlled by antibiotics, there is an increasing pressure of antibiotic resistance [8], and intestinal diseases caused by *C. perfringens* have become more common in farms. In addition, bacterial biofilms act as a shield to protect the bacteria from antibiotics by decreasing their susceptibility to antibiotics, leading to the low efficacy of antibiotics against these biofilm-associated pathogens. These problems are becoming a new and emerging threat to animal agriculture and the development of new antibacterial agents against *C. perfringens* [9]. 

Bacteriophages (phages) are abundant in nature and they are specialized in infecting and killing bacteria [10]. “Phage therapy” is an old idea that has recently come back into vogue [11]. In contrast to antibiotics that kill many types of bacteria, phages kill only one species or strain [12]. Thus, phages can be used to precisely control harmful bacteria and avoid the deleterious effects of broad-spectrum antibiotics on beneficial probiotic bacteria [13,14]. In addition, phages can reduce bacterial populations in biofilms [15]. 

Lytic bacteriophages of *C. perfringens* have been reported previously [16,17]. Although these phages show potential for the lysis of *C. perfringens*, studies relating to the disruption of biofilms formed by this pathogen have not been reported. The current study isolated a novel *C. perfringens* phage DCp1 from the sewage of a donkey farm, sequenced its whole genome, characterized its biological properties, and evaluated its effect on the biofilm of *C. perfringens*. 

## 2. Results

### 2.1. Morphology of Phage DCp1

A new *C. perfringens* phage DCp1 was isolated from the sewage of a donkey farm using *C. perfringens* D22 as a host strain. Phage DCp1 formed a clear plaque (about 1 mm in diameter) on the plate, with a halo of about 3 mm in diameter around the plaque (Figure 1A). The TEM image revealed that phage DCp1 had a regular icosahedral head (about 46 nm in diameter) and a short noncontractile tail (about 40 nm in length) (Figure 1B).

### 2.2. Host Range and EOP

A total of 54 strains were used to determine the host range of phage DCp1 (Appendix A) and *C. perfringens* D22 was used as an indicator bacterium to determine the EOP of phage DCp1. As shown in Appendix A, the lytic activity of DCp1 differed among the tested strains, with a higher EOP indicating that DCp1 had stronger lytic activity in that host strain. Phage DCp1 could only lyse 9.3% (5/54) of *C. perfringens* strains, indicating that phage DCp1 had high host specificity. 

### 2.3. The Optimal MOI and One-Step Growth Curve of Phage DCp1

At the MOIs of 0.01 and 0.1, the phage titer reached the highest value of ~10^10^ PFU/mL (Figure 2A), indicating that the optimal MOI was 0.01~0.1. The one-step growth curve showed that the latent period of phage DCp1 was 25 min, the burst period was 125 min, and the burst size was 85 PFU/cell (Figure 2B).

### 2.4. Thermal and pH Stability of Phage DCp1

For thermal stability, the titers of phage DCp1 showed no significant changes after incubation at 40 °C and 50 °C for 60 min and 60 °C for 40 min. However, the titers of phage DCp1 decreased by 2 orders of magnitude after 60 min of incubation at 60 °C and by 7.5 orders of magnitude after 20 min of incubation at 70 °C, and the phages were completely inactivated after 20 min of incubation at 80 °C (Figure 3A). For pH stability, phage DCp1 was stable over a pH range of 5 to 10 within 3 h. After incubation at pH 4 for 3 h, however, the titers of phage DCp1 decreased by 7 orders of magnitude, and the phages were completely inactivated at pHs 3, 12, and 13. (Figure 3B). The results indicated that phage DCp1 was stable below 60 °C for 1 h over the pH range of 5 to 10.

### 2.5. In Vitro Bactericidal Activity of Phage DCp1

The in vitro bactericidal activity of phage DCp1 against *C. perfringens* D22 is shown in Figure 4. The OD_600_ values of the positive control increased continuously within 24 h, and the OD_600_ values of the negative control remained unchanged. The growth of *C. perfringens* D22 was completely inhibited at all MOIs after treatment with DCp1 during 2–10 h. At 24 h, there was a significant difference in the OD_600_ values (*p* < 0.05) compared with the positive control, and there was no significant difference at different MOIs (*p* > 0.05) (Figure 4A). The bacteria number of D22 remained unchanged during the first 10 hours and only increased by 0.5~2.5 orders of magnitude at 24 h, which was much lower than that in the positive control (Figure 4B). The results demonstrated that DCp1 could significantly inhibit bacterial growth at suitable MOIs and the highest bactericidal activity was found at an MOI of 0.01.

### 2.6. Genomic Characteristics of Phage DCp1

Genomic analysis is an important method for identifying useful functional proteins and the safety of phage applications [18]. Phage DCp1 had a linear double-stranded DNA, with a genome size of 18,555 bp and a G + C content of 28.2%. A total of 25 ORFs were identified in the genome, of which 13 were in the plus strand and 12 were in the minus strand (Figure 5). Only eight ORFs were homologous to genes encoding proteins with known functions, and the others were annotated to encode hypothetical proteins. Five ORFs encoded structural proteins, including bppU family phage baseplate upper protein (ORF7), morphogenesis protein C (ORF8), collar protein (ORF15, ORF16), and tail fibers protein (ORF18). ORF10 encoded DNA polymerase, which was identified as B-type DNA polymerase and involved in DNA replication and modification. It is consistent with the fact that phages encode their own DNA polymerases [19]. ORF14 encoded endolysin, which is related to the release of phage progeny and consists of two domains, the N-acetylmuramoyl-L-alanine amidase catalytic domain (cd02696) and the peptidoglycan-binding domain (pfam05036). Two ORFs showed no similarity in multiple databases. No tRNA, virulence gene, drug resistance gene, and lysogenic gene were found in the genome of phage DCp1. The detailed information on all predicted ORFs was shown in Appendix A. Genomic sequence information and functional annotation of phage DCp1 had been deposited in the GenBank database (accession number: OP256049, submitted on 19 August 2022).

### 2.7. Comparative Analysis

Whole-genome sequence alignment showed that phage DCp1 belonged to the *Guelinviridae*, *Susfortunavirus*, and it had the highest DNA sequence identity (96.95%) with phage vB_CP_qdyz_P5 (accession number: OP894055.1) of the *Susfortunavirus* genus. 

Phage DCp1 has similar genomic features to the phages in Table 1, including short genomes (between 18,000 and 20,000 bp), low GC contents of 28–30%, and no tRNA genes. However, phage DCp1 had very low sequence similarity with these phages (except phage vB_CP_qdyz_P5). In addition, the sequence coverage of phage DCp1 and all phages was very low and there were many unknown genes in the genome of phage DCp1. These results indicated that phage DCp1 might be a novel phage in comparison to the phages in the NCBI database.

### 2.8. Phylogenetic Analysis

The phylogenetic tree based on the whole genome and DNA polymerases indicated that DCp1 was closely related to phage vB_CP_qdzy_P5 and distantly related to the other *C. perfringens* phages, which was consistent with BLASTp analysis (Figure 6A,B). This showed that phage DCp1 was very different from other phages and its novelty was also demonstrated by the taxonomic status. When we further compared the whole genome sequence of DCp1 with six closely related phages, including vB_CP_qdzy_P5, susfortuna, phi24R, LPCPA6, CPS1, and CPD7 using Mauve, we found that phage DCp1 had similar gene module arrangements to most phages (Figure 7). However, the left end (2500–5000 bp) of the DCp1 genome was highly divergent when compared with the other six *C. perfringens* phage genomes and most of the genes in this region had an unknown function.

### 2.9. Effects of Phage DCp1 on the Biofilm of C. perfringens

The SEM image and crystal violet staining showed that *C. perfringens* D22 could form a biofilm and the biofilm reached maturity in 36 h (Figure 8A). To determine the ability of phage DCp1 to inhibit biofilm formation, phage DCp1 was co-cultured with *C. perfringens* D22 at different MOIs. The result showed that phage DCp1 was effective in inhibiting biofilm formation and there was no difference at different MOIs (Figure 8B). Figure 8C and Figure 9 showed that phage DCp1 could completely degrade the biofilm after 5 h of co-cultivation.

## 3. Discussion

*C. perfringens* is one of the most important pathogens in intestinal diseases in newborn foals, causing huge economic losses to the donkey industry [20]. Although no relevant reports have been published, *C. perfringens* has been frequently isolated from diarrheic donkey foals, and diarrhea caused by Clostridial enteritis is becoming more prevalent in donkeys. There are few strategies for preventing and controlling *C. perfringens* infections in donkey farms due to antibiotic resistance. Recently, however, the use of phage therapy to control pathogenic bacteria has been reported [21,22,23]. In this work, we isolated a novel phage of *C. perfringens* and comprehensively analyzed its genomic and biological characteristics. There is a correlation between the burst size and the latency of the phage, as well as the proportion of protein synthesis machinery in the host bacteria [24]. Previous studies have shown that a longer incubation period is associated with a larger burst size [25,26,27]. Compared with phages HN02 and BG3P, phage DCp1 had a slightly longer incubation period and a larger burst size, which was consistent with previous reports [16,17]. Phage DCp1 remained stable at pHs (5~10) for 3 h and temperatures (40 °C~60 °C) within 60 min, indicating that it could survive under various environmental conditions. Although phage DCp1 had a lytic effect on host strain D22, successive passages on the medium showed that phage-resistant mutants still appeared. Based on previous studies, we speculated that phage DCp1 produced phage-resistant mutants at a frequency of about 10^−6^–10^−7^ [28,29].

The halo around the plaque of phage DCp1 may be related to depolymerase [30]. However, no gene that encoded depolymerase was annotated in the genome of DCp1, which requires further analysis. Though phage DCp1 and phage CPS1 were similar in morphology and size, phage DCp1 had relatively low sequence coverage (1%) and sequence similarity (75.13%) with phage CPS1. In addition, the highest sequence coverage of phage DCp1 with known phages in the NCBI database was only 85%. These results indicated that phage DCp1 might be a new phage.

Generally, the phage genome has a modular structure, with each module containing a cluster of genes with a specific function [31]. ORF7 and ORF18 encoded the upper baseplate protein (BppU) and tail fiber protein of phage DCp1, respectively. BppU is composed of six asymmetric trimers which link to the Dit central core and the receptor binding proteins (RBPs) [32]. The location of RBPs varies among phages. For example, phage T4 is attached to host bacteria by tail fibers, while phage TP901-1 has RBPs in the baseplate [33,34]. Therefore, further studies are necessary to confirm the RBPs of phage DCp1. ORF14 encoded endolysin, which can degrade peptidoglycan in host cell walls to inhibit or kill bacteria [35]. Endolysin of DCp1 consists of two domains, including N-acetylmuramoyl-L-alanine amidase catalytic domain and peptidoglycan-binding domain. N-acetylmuramoyl-L-alanine amidase (also known as peptidoglycan aminohydrolase) is an autolysin. Enzymes containing this domain can degrade the peptidoglycan by cleaving the amide bond between N-acetylmuramoyl and L-amino acids [36,37]. The phages that infect Gram-positive bacteria are usually host-specific, but their endolysins have a broad host spectrum [38,39]. Further research is needed to determine if this is the case for phage DCp1. In general, endolysins have to pass through the pores in the cell membrane created by holin [40]. However, the holin gene was not identified in the DCp1 genome and holin may also be among the hypothetical proteins or the unannotated proteins. The B-type DNA polymerase encoded by ORF10 had both an obligatory 5′→3′ DNA synthesis activity and an optional exonuclease activity [41]. However, most B-type DNA polymerases lack the domain that is required for 3′→5′ exonuclease [42]. ORF16 was predicted to encode a hypothetical protein that contained a phage connector domain and served as an interface for tail attachment and binding [43]. Notably, holin was not identified in the genome of phage DCp1. The holin gene is most likely downstream of the lysin gene. This placement is unique to the other clostridial bacteriophages [44]. Based on publicly available genomic information at NCBI, there are only four phages in the genus *Susfortunavirus*. Therefore, the characterization of phage DCp1 is important in expanding our knowledge of *C. perfringens* phages. 

Biofilm is a dominant organizational form of bacterial life in nature. Due to the presence of biofilms, bacteria can increase antibiotic resistance, protect themselves from phagocytosis, and resist physical and environmental stresses [45,46]. Previous studies have confirmed that phages have a scavenging effect on bacterial biofilms. For example, phages vB_SauM-A, vB_SauM-C, and vB_SauM-D can eradicate the biofilm of *Staphylococcus aureus* [47]. Phage MJ2 is effective against the biofilm of *Enterobacter cloacae* [48]. Phage AB7-IBB1 has a scavenging effect on *Acinetobacter baumannii* biofilm [49]. *C. perfringens* can form biofilms; however, no studies have been reported on the effects of phages on *C. perfringens* biofilms. In this study, the results demonstrated that phage DCp1 had high antibacterial efficacy against *C. perfringens* biofilms and most of the biofilms were removed after treatment with DCp1 for 5 h, indicating that phage DCp1 could be used as an effective antibacterial agent against *C. perfringens* biofilms. 

## 4. Materials and Methods

### 4.1. Strains and Conditions

A total of 54 *C. perfringens* strains were used in this work (Appendix A), which were previously identified and stored in the Veterinary Microbiology Laboratory of Qingdao Agricultural University. *C. perfringens* strains were cultured anaerobically at 37 °C in the TSC Agar and Anaerobic Meat Liver (AML) broth (Qingdao Haibo Biotechnology Co., Ltd., Qindgao, China) in an anaerobic incubator (Shanghai Jiehan Laboratory Equipment Co., Ltd., Shanghai, China). Antibiotic susceptibility of the strains was determined by the disk diffusion method (Appendix A) [50].

### 4.2. Sample Enrichment and Bacteriophage Isolation

Sewage samples were collected from donkey farms in Shandong, China. Phages were isolated from samples by the double-layer agar method using *C. perfringens* D22 as the host bacterium [24]. After samples were centrifuged at 12,000 r/min for 5 min, the supernatant was filtered through a 0.22 μm filter, mixed with fresh AML broth (10^9^ CFU/mL, 100 μL) and incubated overnight at 37 °C. The enrichment process was repeated three times to obtain the final filtrate. A volume (100 μL) of the filtrate was mixed with 100 μL of *C. perfringens* suspension (10^9^ CFU/mL) and incubated at 37 °C for 5 min. The mixture was added to 5 mL of 0.7% molten agar and poured onto the prepared Nutrient Agar (NA) plate. Plates were incubated anaerobically overnight at 37 °C and single plaques were selected and purified several times until the shape and size of plaques were uniform, and the titer of purified phage was determined by the double-layer agar method.

### 4.3. Electron Microscopy

Transmission electron microscopy (TEM) was used to observe the morphology of phage DCp1 [51]. In brief, 20 μL of phage suspension (10^9^ PFU/mL) was placed on a carbon-coated grid and allowed to be adsorbed for 15 min, followed by staining with 1% phosphotungstic acid for 5 min. The grid was air-dried at 70 °C in the dark. The morphology of phage DCp1 was examined using a transmission electron microscope (TEM, Hitachi, Tokyo, Japan) at an accelerated voltage of 80 kV.

### 4.4. Host Range and Efficiency of Plating (EOP) Measurement

The host range of phage DCp1 was determined by a spot test [52]. The lytic activity of phage DCp1 against different strains of *C. perfringens* was determined using a spot test (Appendix A). Briefly, *C. perfringens* suspension (10^9^ CFU/mL, 100 μL) was mixed with 0.7% molten agar (5 mL) and poured onto the prepared nutrient agar (NA) plates. After solidification, 5 μL of phage suspension (10^9^ PFU/mL) was dropped on the top of the NA plate. After complete absorption, the plate was cultured overnight at 37 °C to observe the formation of plaques. The titer of phage DCp1 against the spotted strain was determined by the double-layer agar method. The EOP was determined by the ratio of PFUs of phage DCp1 from each susceptible strain to CFUs from the indicator strain of *C. perfringens* D22 [53]. Each experiment was repeated three times.

### 4.5. Optimal Multiplicity of Infection (MOI)

The optimal MOI of phage DCp1 was determined as described previously [54]. Firstly, phage suspension was mixed with fresh *C. perfringens* D22 suspension (10^8^ CFU/mL) at different ratios of 10, 1, 0.1, 0.01, and 0.001. Then, the mixture was incubated at 37 °C for 3 h, followed by centrifugation at 12,000 r/min for 30 s. The supernatant was collected for the determination of the phage titers using the double-layer agar method. The ratio with the highest phage titer was considered the optimal MOI. 

### 4.6. One-Step Growth Curve

The one-step growth curve of phage DCp1 was determined as described previously [55]. Briefly, phage suspension (10^9^ PFU/mL) was mixed with fresh D22 suspension at an MOI of 10 and incubated at 37 °C for 5 min. The mixture was centrifuged at 12,000 r/min for 3 min and the pellet was washed twice with AML broth to remove unabsorbed phages, resuspended in AML broth, and incubated at 37 °C with gentle shaking. Aliquots (200 μL) of the culture were taken at 5-min intervals in the first hour, 20-min intervals in the second hour, and 30-min intervals in the third hour. The aliquots were centrifuged at 12,000 r/min for 5 min and the supernatants were collected for the determination of phage titers using the double-layer plate method. The burst size was calculated as the ratio of the final count of liberated phage particles to the initial phage particles.

### 4.7. Thermal and pH Stability

Thermal and pH stability of phage DCp1 were determined as described previously [56]. For thermal stability, the phage suspension was incubated at 40 °C, 50 °C, 60 °C, 70 °C, and 80 °C and aliquots were taken at 20 min, 40 min, and 60 min for titer determination. For pH stability, the phage suspension was incubated over a range of pHs (3–13) at 37 °C for 1, 2, and 3 h. Phage titers were determined using the double-layer agar method. 

### 4.8. In Vitro Bactericidal Activity

The bactericidal activity of DCp1 against the host strain *C. perfringens* D22 was assessed using optical densitometry and bacterial colony counting [21]. Briefly, DCp1 was cultured with *C. perfringens* D22 (10^8^ CFU/mL) in AML broth at various MOIs (10, 1, 0.1, 0.01, and 0.001), followed by incubation at 37 °C. A UV-vis spectrophotometer was used to measure the optical density (OD) at 600 nm in a 96-well plate at 1 h intervals during the first 10 h and at 24 h. The bacterial growth was also monitored by measuring bacterial titers at two-hour intervals during the first 10 h and at 24 h. A bacterial culture without phages served as a positive control and AML broth without bacteria and phages was used as a negative control. Each aliquot was measured in triplicate.

### 4.9. Genome Extraction, Sequencing, and Bioinformatics Analysis

The genomic DNA of phage DCp1 was extracted using a viral genomic DNA/RNA extraction kit (Tiangen Biochemical Technology Co., Ltd., Beijing, China). Sequencing was performed by Huitong Biotechnology Co., Ltd. (Shenzhen, China). The purified genomic DNA was sheared into c. 350 bp fragments to construct a paired-end (PE) library using a Nextera XT sample preparation kit (Illumina, San Diego, CA, United States). The PE reads of 150 bp were generated using a Novaseq 6000 sequencer (Illumina, San Diego, CA, United States). High-quality reads were assembled into the phage genome using the de novo assembler SPAdes v.3.11.0 [57]. The complete sequence of phage DCp1 was annotated using RAST (http://rast.nmpdr.org) and GeneMark (http://opal.biology.gatech.edu/Gene Mark/) (accessed on 10 August 2022) [58,59]. Predicted ORFs were verified using online BLASTP (http://www.ncbi.nlm.nih.gov/BLAST) (accessed on 1 February 2023). Putative transfer RNA (tRNA)-encoding genes were searched using tRNAscan-SE (http://trna.ucsc.edu/tRNAscan-SE/)(accessed on 10 August 2022) [60]. SnAp Gene was used to construct the whole genome map and MEGA5 was used to construct the phylogenetic tree based on the whole-genome sequence and DNA polymerases [61]. Comparisons of complete genome sequences between phage DCp1 and other phages were performed using Mauve [62].

### 4.10. Effects of Phage DCp1 on C. perfringens Biofilm

#### 4.10.1. Biofilm Assay

The biofilm of *C. perfringens* D22 was prepared by previously described methods with some modifications [49,63]. Briefly, *C. perfringens* cultures were diluted in fresh Thioglycollate medium (Qingdao Haibo Biotechnology Co., Ltd., Qingdao, China) at a final concentration of 10^6^ CFU/mL, and 200 μL diluted bacteria was added to each well of the 24-well plates, followed by incubation at 37 °C for 24 h, 36 h, and 48 h. The blank control was the medium without *C. perfringens* D22. After incubation, the wells were washed with sterile phosphate-buffered saline (PBS) twice and fixed with methanol for 15 min. After air-drying, the plates were stained with 2% crystal violet and kept at room temperature for 15 min. After the removal of excessive stain and washing with PBS, 33% glacial acetic acid was added to dissolve the stain. Absorbance was recorded at OD_570_ nm. 

#### 4.10.2. Inhibitory Effect of Phage DCp1 on Biofilm Formation

The ability of phage DCp1 to inhibit biofilm formation was analyzed according to the previously described method [64]. Overnight, *C. perfringens* cultures were diluted in the fresh Thioglycollate medium up to 10^6^ CFU/mL. The diluted *C. perfringens* culture (200 μL) was mixed with 200 μL of phage suspension at different MOIs (1, 0.1, 0.01, and 0.001) and incubated in 24-well cell culture plates at 37 °C for 36 h. The blank control was the medium without phages and *C. perfringens*. Biofilm formation was assessed using crystal violet staining. 

#### 4.10.3. Eradication of *C. perfringens* Biofilms by Phage DCp1

The ability of phage DCp1 to eradicate the biofilms was assessed according to the previous method [47]. Biofilms were grown in 24-well cell culture plates as described above and rinsed with PBS. Phage suspension (10^9^ PFU/mL, 200 μL) was added and incubated at 37 °C for 5 h. Biofilms were assessed using crystal violet staining. 

The morphology of *C. perfringens* biofilms treated with phage DCp1 was observed using scanning electron microscopy [65]. The biofilms were grown on coverslips and treated as described above. After treatment, the biofilms were fixed with 3% glutaraldehyde for 4 h, washed gently with PBS (10 mM, pH 7.4), and fixed with 1% osmic acid for 1.5 h. The biofilms were then washed gently with 10 mM PBS, dehydrated using a graded ethanol series (30%, 50%, 70%, and 80% once for 10 min, and 100% twice for 10 min each time), and displaced by isoamyl acetate lipid (50% once, 100% twice). The biofilms were dried using a critical point dryer, coated with gold, and photographed on a scanning electron microscope (SEM, Hitachi, Tokyo, Japan).

### 4.11. Statistical Analysis

All experiments were performed at least three times and three biological repeats were performed for each experiment. The results were statistically analyzed using GraphPad Prism (version 6.02, GraphPad Software, La Jolla, CA, USA) and the data were analyzed by one-way analysis of variance to compare significant differences. Statistical significance was set at *p* < 0.05.

## 5. Conclusions

The current study described a novel *C. perfringens* phage DCp1 isolated from the sewage of a donkey farm. Whole-genome sequencing showed that phage DCp1 had a linear double-stranded DNA genome with a size of 18,555 bp and a G + C content of 28.2%. Phylogenetic analysis showed that phage DCp1 belonged to the family *Guelinviridae*, *Susfortunavirus* and had a very low sequence identity with the known phage sequences in the NCBI database. No tRNA, virulence gene, drug resistance gene, and lysogenic gene were identified in the genome of phage DCp1. The high host specificity of phage DCp1 enabled it to be used for more precise sterilization. Moreover, the advantages, such as the large burst size, high thermal and pH stability, and high efficacy of biofilm removal can broaden the practical application of phage DCp1 in bacterial control.

## Figures and Tables

**Figure 1 ijms-24-04191-f001:**
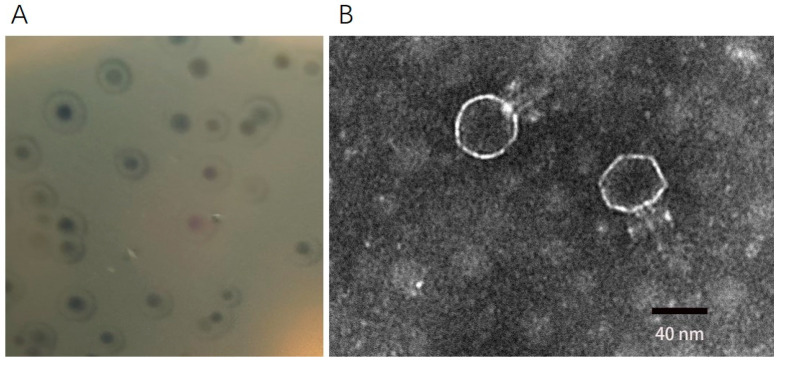
Morphology of phage DCp1. (**A**) Plaques produced by phage DCp1 on the lawn of *C. perfringens*. (**B**) TEM image of phage DCp1 with a regular icosahedral head (46 nm in diameter) and a short tail (40 nm in length).

**Figure 2 ijms-24-04191-f002:**
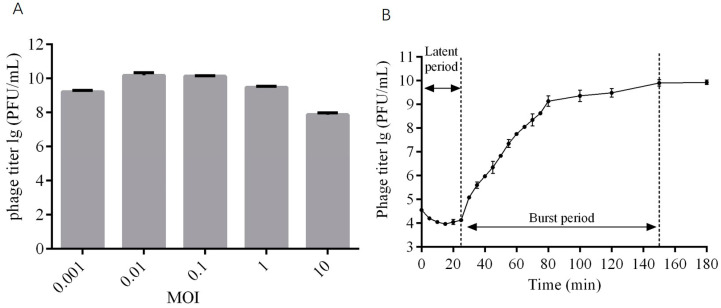
The optimal MOI and one-step growth curve of phage DCp1. (**A**) The titers of phage DCp1 at different MOIs (0.001, 0.01, 0.1, 1, and 10). (**B**) The one-step growth curve of phage DCp1. Phage DCp1 was mixed with *C. perfringens* D22 at the MOI of 10. The titer of phage DCp1 was determined at different time points within 3 h.

**Figure 3 ijms-24-04191-f003:**
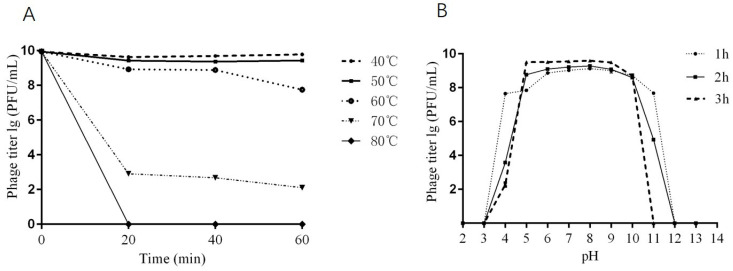
Thermal and pH stability of phage DCp1. (**A**) The titers of phage DCp1 at different temperatures (40 °C, 50 °C, 60 °C, 70 °C, and 80 °C) and different time points (20, 40, and 60 min). (**B**) The titers of phage DCp1 at different pHs (3~13) and different time points (1, 2, and 3 h).

**Figure 4 ijms-24-04191-f004:**
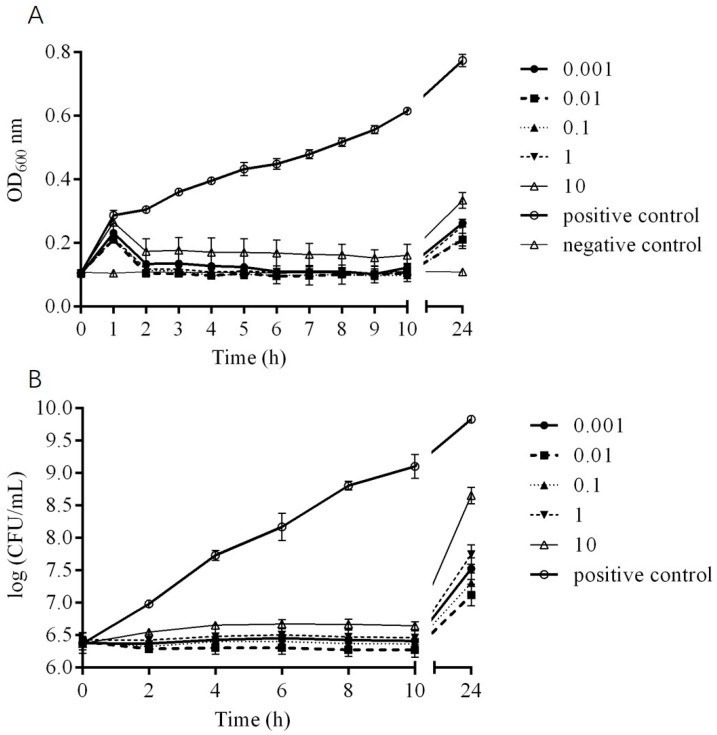
In vitro bactericidal activity of phage DCp1. (**A**) In vitro bactericidal activity of phage DCp1 against *C. perfringens* D22 strain at different MOIs (10, 1, 0.1, 0.01, and 0.001). (**B**) CFUs of phage DCp1 against *C. perfringens* D22 at different MOIs (10, 1, 0.1, 0.01, and 0.001). The data are expressed as means ± SD (n = 3).

**Figure 5 ijms-24-04191-f005:**
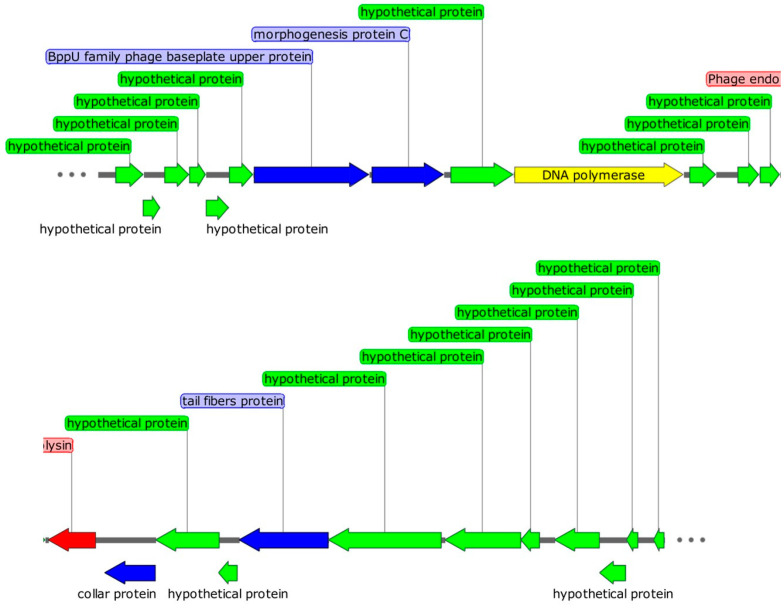
Genome map of phage DCp1. Green arrows represent the genes encoding the hypothetical proteins; yellow arrows represent the genes encoding DNA replication and modification proteins; blue arrows represent the genes encoding structural and packaging proteins; the red arrows represent the genes encoding host lytic proteins.

**Figure 6 ijms-24-04191-f006:**
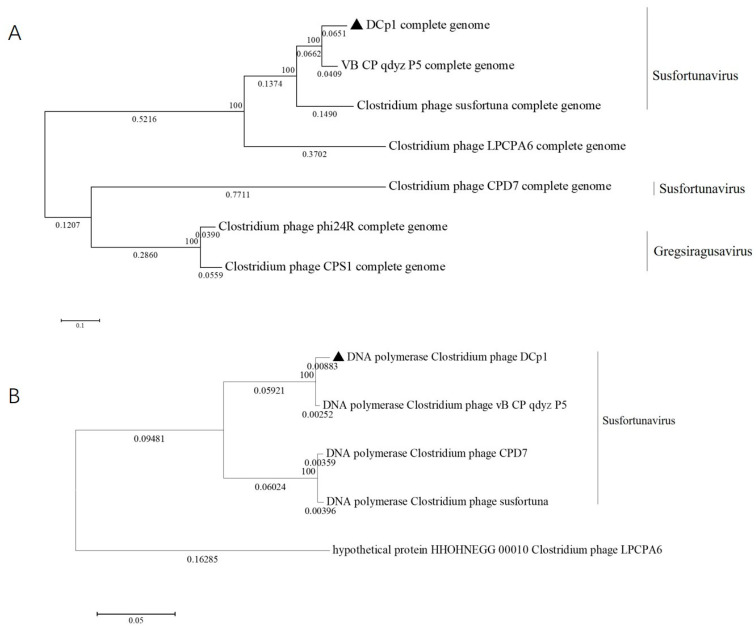
Phylogenetic trees showing the relationship between phage DCP1 and other phages in the NCBI database. (**A**) Phylogenetic tree based on the whole genome. (**B**) Phylogenetic tree based on DNA polymerases.

**Figure 7 ijms-24-04191-f007:**
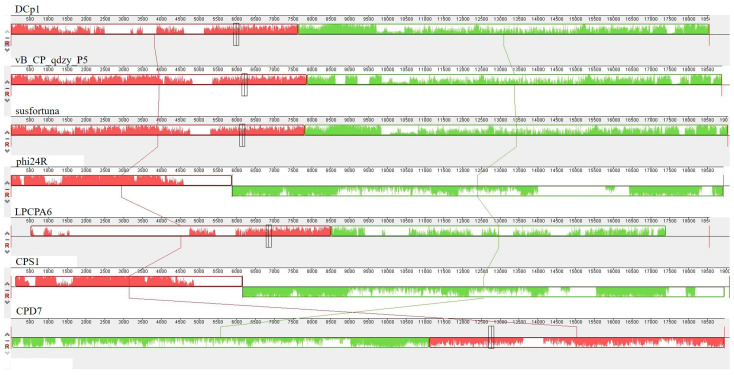
Comparative genomic analysis of phage DCp1 with phages vB_CP_qdzy_P5, susfortuna, phi24R, LPCPA6, CPS1, and CPD7 using Mauve. The nucleotide sequence similarity is indicated by the height of the colored bars, while the regions that are dissimilar are shown in white.

**Figure 8 ijms-24-04191-f008:**
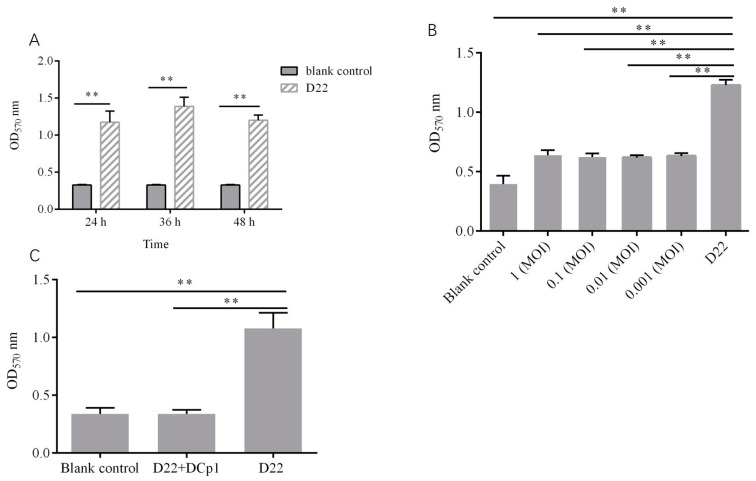
Effects of phage DCp1 on the biofilm of *C. perfringens* D22. **: *p* < 0.01 (**A**) Biofilm formation of *C. perfringens* D22 after different periods of incubation. (**B**) Inhibitory effect of phage DCp1 on the biofilm at different MOIs. (**C**) Eradicative effect of phage DCp1 on the biofilm. Phage DCp1 completely eradicated the biofilm after 5 h.

**Figure 9 ijms-24-04191-f009:**
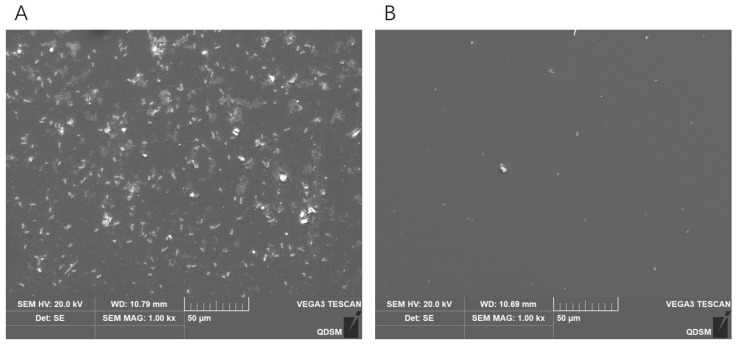
Eradication of *C. perfringens* D22 biofilms by phage DCp1. (**A**) SEM image of untreated *C. perfringens* D22 biofilm. (**B**) SEM image of phage DCp1-treated *C. perfringens* D22 biofilm.

**Table 1 ijms-24-04191-t001:** Comparison of general genomic features of phage DCp1 and other closely related phages.

Genome Characteristics	Phage						
	DCp1	vB_CP_ qdyz_P5	Susfortuna	CPD7	phi24R	CPS1	LPCPA6
Genome size (bp)	18,555	18,888	19,046	18,958	18,919	19,089	18,554
G + C (%)	28.20	28.80	29.22	29.12	27.80	28.26	30.56
Predicted ORFs	25	27	27	26	21	26	25
tRNAs	0	0	0	0	0	0	0
Similarity with DCp1 (%)	100	96.95	86.23	86.04	77.64	75.13	73.63
Coverage with DCp1 (%)	100	85	74	74	0	1	21
Accession no.	OP256049	OP894055.1	NC_048712	MK017820	JN800508	NC_048661	OM638104

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
