# Peer review of "Characterization and Genomic Analysis of a Novel Lytic Phage DCp1 against Clostridium perfringens Biofilms"

_ijms, 2023, doi:10.3390/ijms24044191_

Round 1

Reviewer 1 Report

In this study, authors isolated Phage DCp1 specific to C. perfrinegens strains.  They characterized the isolated phage using physical and genomic methods. They evaluated the ability of the DCp1 to inhibit and destroy the biofilms of C. perfringens stains.

Comments:

1- figure 1 must be provided at a better quality.

2- What is the antibiotic resistance and virulence profile for the C.perfringens strains used in this study?

The antibiotic sensitivity test must be done and included in the manuscript.

3- The results of the host range did not described adequately.

4- The DCp1 phage infected only 9.3% of the strains. Despite the phage showing specificity to the host, the phage has a narrow host range, which is not preferable in therapy. In this case, only one phage is not enough, and authors still need to use a cocktail of phages to widen the phage host range.  

5-Authors mentioned that they determined the host range by plaque assay, but the method used seemed like a spot assay. They did not make a serial dilution of the phage stock and mixed it with the host in the top agar and poured it on the solid agar plate. They only spotted phage suspension from the phage stock on the plates containing the host and this is referred to spot test, not a plaque assay.

6- The Efficiency of Plating (EOP) test must be performed for strains sensitive to phage infection to show phage productivity. The EOP is a quantitative and not only qualitative measure of phage host range.

7- Results corresponding to the temperature and pH stability are very limited and lacking in detail. Please, describe more details about quantitative and statistical analysis.

8- what is the frequency of the phage-resistant mutants for C.perfrinegens treated with DCp1 phage?

9- The authors did not show the effect of the DCp1 phage on the planktonic cells. This experiment should be done and included in the manuscript.

10- Authors mentioned five phages as the closest phage genomes to the DCp1 phage. Why did they not include Clostridium phage vB_CP_qdyz_P5 (OP894055.1) which showed 96.95% identity and 83 % coverage to isolated phage?

11- A figure with the alignment of the DCp1 phage with the other C.perfringens virulent phages should be done using bioinformatics tools and included in the manuscript, and the differences should be discussed.

12- What is new about this phage? Please, describe the unique features of the phage instead of just telling the reader to look at the table in the supplementary.

13- In the biofilm assay, the authors evaluate the biofilm inhibition and distraction only by the OD measurements and SEM. In addition to that, the authors should quantify the bacteria after treatment by quantification methods such as the CFU method to determine the viable count.

14-The different MOI should be used in the biofilm assay not only one in order to show the efficiency of phage at low and high MOI.

15- The application experiments must be done. The authors should design an experimental model to show the potential application of isolated phage.

Author Response

Dear Reviewer:

We greatly appreciated for your valuable suggestions and comments on our manuscript. We have carefully revised the manuscript according to these comments and suggestions.

The responses to your comments are given as follows.

Point 1:  Figure 1 must be provided at a better quality.

Response 1: We agree with your comments. However, the short tail of phage DCp1 is only 40 nm in length, which is difficult to be photographed clearly under the electron microscope. This can also be confirmed by electron microscope images of other similar phages[1-4].

  1. Tian, Y.; Wu, L.; Lu, R.; Bao, H.; Zhou, Y.; Pang, M.; Brown, J.; Wang, J.; Wang, R.; Zhang, H. Virulent phage vB_CpeP_HN02 inhibits Clostridium perfringens on the surface of the chicken meat. Int J Food Microbiol 2022, 363, 109514, doi:10.1016/j.ijfoodmicro.2021.109514.
  2. Ha, E.; Chun, J.; Kim, M.; Ryu, S. Capsular Polysaccharide Is a Receptor of a Clostridium perfringens Bacteriophage CPS1. Viruses 2019, 11, doi:10.3390/v11111002.
  3. Ha, E.; Son, B.; Ryu, S. Clostridium perfringens Virulent Bacteriophage CPS2 and Its Thermostable Endolysin LysCPS2. Viruses 2018, 10, doi:10.3390/v10050251.
  4. Volozhantsev, N.V.; Oakley, B.B.; Morales, C.A.; Verevkin, V.V.; Bannov, V.A.; Krasilnikova, V.M.; Popova, A.V.; Zhilenkov, E.L.; Garrish, J.K.; Schegg, K.M.; et al. Molecular characterization of podoviral bacteriophages virulent for Clostridium perfringens and their comparison with members of the Picovirinae. PLoS One 2012, 7, e38283, doi:10.1371/journal.pone.0038283.

Point 2: What is the antibiotic resistance and virulence profile for the C.perfringens strains used in this study? The antibiotic sensitivity test must be done and included in the manuscript.

Response 2: According to your suggestion, the information on antibiotic resistance in each strain has been supplemented in Table S2 (Lines 269-270). In this work, however, we did not want to discuss virulence profile as it was quite different, and we will further investigate it with a cocktail of different phages and relevant experimental models, which will be discussed in other papers.

Point 3: The results of the host range did not described adequately.

Response 3: According to your suggestion, we have supplemented the manuscript with a description of the host range (Lines 68-74).

Point 5: Authors mentioned that they determined the host range by plaque assay, but the method used seemed like a spot assay. They did not make a serial dilution of the phage stock and mixed it with the host in the top agar and poured it on the solid agar plate. They only spotted phage suspension from the phage stock on the plates containing the host and this is referred to spot test, not a plaque assay.

Response 5: According to your suggestion, we have made corresponding changes in the revised manuscript. The plaque assay was changed to the spot test and double-layer agar method was supplemented (Lines 292-301).

Point 6: The Efficiency of Plating (EOP) test must be performed for strains sensitive to phage infection to show phage productivity. The EOP is a quantitative and not only qualitative measure of phage host range.

Response 6: According to your suggestion, we have made corresponding supplements in the revised manuscript (Table S1, Lines 68-74, Lines 298-301).

Point 7: Results corresponding to the temperature and pH stability are very limited and lacking in detail. Please, describe more details about quantitative and statistical analysis.

Response 7: According to your suggestion, we have supplemented the manuscript with a description of the results corresponding to the temperature and pH stability (Lines 88- 96).

Point 8: What is the frequency of the phage-resistant mutants for C. perfrinegens treated with DCp1 phage?

Response 8: Your comment is greatly appreciated. Based on previous studies, we speculated that phage DCp1 produces phage-resistant mutants at a frequency of about 10-6-10-7. And we have already mentioned in the discussion (Lines 211-215).

Point 9: The authors did not show the effect of the DCp1 phage on the planktonic cells. This experiment should be done and included in the manuscript.

Response 9: According to your suggestion, we have supplemented the in vitro bactericidal activity in the manuscript (Lines 101-112, Lines 327-336).

Point 10: Authors mentioned five phages as the closest phage genomes to the DCp1 phage. Why did they not include Clostridium phage vB_CP_qdyz_P5 (OP894055.1) which showed 96.95% identity and 83% coverage to isolated phage?

Response 10: Your comment is greatly appreciated. The sequence submission date for phage DCp1 was August 19, 2022, while the submission date for phage vB_CP_qdyz_P5 was December 2022. Therefore, the information on phage vB_CP_qdyz_P5 was not available in the GenBank database when we performed the genomic analysis. Nevertheless, according to your suggestion, we have made corresponding supplements and changes in the revised manuscript (Lines 136-137, Lines 145-154, Lines 158-167, Table 1).

Point 11: A figure with the alignment of the DCp1 phage with the other C. perfringens virulent phages should be done using bioinformatics tools and included in the manuscript, and the differences should be discussed.

Response 11: According to your suggestion, we have supplemented a figure with the alignment of the DCp1 phage with the other C. perfringens virulent phages in the revised manuscript (Figure 7, Lines 162-167).

Point 12: What is new about this phage? Please, describe the unique features of the phage instead of just telling the reader to look at the table in the supplementary.

Response 12: According to your suggestion, we have made corresponding supplements in the revised manuscript. The features of phage DCp1 were supplemented (Line 148-154).

Point 13:  In the biofilm assay, the authors evaluate the biofilm inhibition and distraction only by the OD measurements and SEM. In addition to that, the authors should quantify the bacteria after treatment by quantification methods such as the CFU method to determine the viable count.

Response 13: We agree with your suggestions. This is a limitation of this study. The purpose of this assay was to investigate the effects of phage DCp1 on the biofilm, so we did not focus on the effect of phage DCp1 on individual bacteria in the biofilm. Given your valuable suggestion, we’ll further address these details in the future.

Point 14: The different MOI should be used in the biofilm assay not only one in order to show the efficiency of phage at low and high MOI.

Response 14: According to your suggestion, we repeated the experiment with different MOIs (Line 369-371, Line 182-185).

Point 4: The DCp1 phage infected only 9.3% of the strains. Despite the phage showing specificity to the host, the phage has a narrow host range, which is not preferable in therapy. In this case, only one phage is not enough, and authors still need to use a cocktail of phages to widen the phage host range.

Point 15: The application experiments must be done. The authors should design an experimental model to show the potential application of isolated phage.

Response 4 and 15: We agree with your suggestions. Although the host bacterium C. perfringens D22 was isolated from a donkey with diarrhea, it is not necessarily the only causative agent. In most cases, C. perfringens is an opportunistic pathogen. We have previously measured the pathogenicity of C. perfringens D22. Its pathogenicity is somewhat less than satisfactory. Only 30% (3/10) of Galleria mellonella were killed by 10 μL of D22 bacterial solution (109 CFU/mL). DCp1 has a narrow host spectrum and is not advisable in therapy alone. Therefore, we’re planning to further combine phage DCp1 with other phages for animal experiments in the future, and we will discuss this in detail in another paper.

Reviewer 2 Report

The manuscript on Characterization and Genomic Analysis of a Novel Lytic Phage 2DCp1 against Clostridium perfringens Biofilms describes characteristics of novel phage of great application potential. The manuscript is a comprehensive description of novel phage regarding its genome sequence, phage particle morphology and its properties (effects of phage DCp1 on the biofilm of C. perfringens). The manuscript is scientifically sound and well written, easy to read and understand. The methods used are well chosen and allowed to obtain a complete description of the phage I strongly recommend to publish the manuscript only with minor editorial corrections (e.g. lines 317 and 324 – instead of “previous method” I suggest “previously described method”).

Author Response

Dear Reviewer:

Thank you very much for your valuable suggestions and comments on our manuscript. We have carefully revised the manuscript according to these comments and suggestions.

The responses to your comments are given as follows.

Point 1: The methods used are well chosen and allowed to obtain a complete description of the phage I strongly recommend to publish the manuscript only with minor editorial corrections (e. g. lines 317 and 324 – instead of “previous method” I suggest “previously described method”).

Response 1: According to your suggestion, we have changed “previous method” to “previously described method” (Line 356 and Line 368).

Reviewer 3 Report

This is a clearly written and enjoyable short paper describing a novel lytic phage. I have only a few points to add or change which would improve the quality of the paper.

Fig. 2B: This experiment is rather a 2-step (or more steps) growth, which usually happens at a low MOI. The first step is probably at about 25 min (as stated by the authors) and the second step at about 50 min p.I.  To do an exact one-step growth, it is better to infect with an MOI of 10 and follow the OD of the culture every five min and in parallel the released progeny. Then you obtain a clear view of the cell lysis and the phage burst.

Line 278: It should read: The aliquots (not the culture) were centrifuged at..

Fig1S: Here it is not clear what the authors mean by "lytic activity". As it is described they provide rather an estimation of EOP (efficieny of plating). Otherwise the OD of a culture should be estimated after phage infection to see if the phage causes lysis or growth arrest.

Author Response

Dear Reviewer:

Thank you very much for your valuable suggestions and comments on our manuscript. We have carefully revised the manuscript according to these comments and suggestions.

The responses to your comments are given as follows.

Point 1: Fig. 2B: This experiment is rather a 2-step (or more steps) growth, which usually happens at a low MOI. The first step is probably at about 25 min (as stated by the authors) and the second step at about 50 min p. I. To do an exact one-step growth, it is better to infect with an MOI of 10 and follow the OD of the culture every five min and in parallel the released progeny. Then you obtain a clear view of the cell lysis and the phage burst.

Response 1: We agree with your comments. According to your suggestion, one-step growth curve was repeated with an MOI of 10 (Lines 77-79 and Figure 2B).

Point 2: Line 278: It should read: The aliquots (not the culture) were centrifuged at.

Response 2: According to your suggestion, we have changed “the culture” to “the aliquots” (Line 316).

Point 3: Fig1S: Here it is not clear what the authors mean by "lytic activity". As it is described they provide rather an estimation of EOP (efficiency of plating). Otherwise the OD of a culture should be estimated after phage infection to see if the phage causes lysis or growth arrest.

Response 3: According to your suggestion, we have added to the test about EOP (Table S1, Line 68-74, Line 298-301).

Round 2

Reviewer 1 Report

Thank you to the authors for their valuable responses and improvements in the manuscript. I recommend accepting the paper for publication.

Best regards